# The Genetics of Polycystic Ovary Syndrome: An Overview of Candidate Gene Systematic Reviews and Genome-Wide Association Studies

**DOI:** 10.3390/jcm8101606

**Published:** 2019-10-03

**Authors:** Danielle Hiam, Alba Moreno-Asso, Helena J. Teede, Joop S.E. Laven, Nigel K. Stepto, Lisa J. Moran, Melanie Gibson-Helm

**Affiliations:** 1Institute of Health and Sport, Victoria University, Melbourne 3011, Australia; danielle.hiam@vu.edu.au (D.H.); alba.moreno@vu.edu.au (A.M.-A.); nigel.stepto@vu.edu.au (N.K.S.); 2Monash Centre for Health Research and Implementation, School of Public Health and Preventive Medicine, Monash University, Melbourne 3168, Australia; helena.teede@monash.edu (H.J.T.); lisa.moran@vu.edu.au (L.J.M.); 3Division of Reproductive Endocrinology and Infertility, Department of Obstetrics and Gynaecology, Erasmus University Medical Center, 3000 CA Rotterdam, The Netherlands; j.laven@erasmusmc.nl; 4Australian Institute of Musculoskeletal Science (AIMSS), Victoria University, St Albans 3021, Australia

**Keywords:** Polycystic Ovary Syndrome, genetics, genetic association studies, Systematic Review, Overview of Systematic Reviews

## Abstract

Polycystic Ovary Syndrome (PCOS) is a complex condition with mechanisms likely to involve the interaction between genetics and lifestyle. Familial clustering of PCOS symptoms is well documented, providing evidence for a genetic contribution to the condition. This overview aims firstly to systematically summarise the current literature surrounding genetics and PCOS, and secondly, to assess the methodological quality of current systematic reviews and identify limitations. Four databases were searched to identify candidate gene systematic reviews, and quality was assessed with the AMSTAR tool. Genome-wide association studies (GWAS) were identified by a semi structured literature search. Of the candidate gene systematic reviews, 17 were of high to moderate quality and four were of low quality. A total of 19 gene loci have been associated with risk of PCOS in GWAS, and 11 of these have been replicated across two different ancestries. Gene loci were located in the neuroendocrine, metabolic, and reproductive pathways. Overall, the gene loci with the most robust findings were *THADA, FSHR, INS-VNTR*, and *DENND1A*, that now require validation. This overview also identified limitations of the current literature and important methodological considerations for future genetic studies. Much work remains to identify causal variants and functional relevance of genes associated with PCOS.

## 1. Introduction

Polycystic ovary syndrome (PCOS) is a major public health concern affecting 6–10% of reproductive aged women [1]. PCOS is exacerbated by obesity and has significant metabolic, reproductive, and psychological features, including an increased risk of Type 2 Diabetes Mellitus (T2DM) with an earlier age of onset, subfertility, and an increased risk of depression and anxiety symptoms [2,3,4]. At present, the internationally accepted criteria for diagnosis of PCOS is the revised Rotterdam criteria [5], which requires exclusion of other causes of hyperandrogenism, like adrenal or pituitary dysfunction, and presence of two of the following three characteristics: oligo- or anovulation, clinical and/or biochemical signs of hyperandrogenism, and polycystic ovaries on ultrasound. The Rotterdam criteria yields four phenotypes of PCOS, and there is evidence that the different PCOS phenotypes have varying degrees of adiposity, and may differ in metabolic and reproductive profiles [6]. The proposed pathophysiology of PCOS is a synergistic relationship between perturbed gonadotrophin releasing hormones (GnRH) pulsatility and hyperandrogenism probably accompanied by hyperinsulinemia, insulin resistance, and inflammation. However, the nuances of these relationships are yet to be fully elucidated (Figure 1) [7,8,9].

In addition, PCOS pathophysiology appears to have a polygenic predisposition that is exacerbated by environmental factors, especially obesity [2]. The polygenic predisposition is well documented with familial clustering of PCOS symptoms, providing evidence for a genetic contribution to the pathophysiology [10]. Both female and male family members of women diagnosed with PCOS share common characteristics of the syndrome. Moreover, they seem to be more prone to develop T2DM and metabolic syndrome [11]. Finally, mono- and dizygotic twin studies have demonstrated the heritability of PCOS to be approximately 70% [12].

Two complementary types of genetic studies are Genome Wide Association Studies (GWAS) and candidate gene studies. GWAS look for associations between common genetic polymorphisms and the trait or disease without a predefined hypothesis about the possible role of genetic variants in the pathophysiology. However, a common misunderstanding about GWAS is that they identify specific genes. They merely provide information about a genetic region (gene loci) that is significantly associated with the trait. On one hand, the identified gene loci might be directly involved in gene function if located in or near a gene, or they might have a regulatory function for genes up- or downstream. Hence, genetic loci detected by GWAS provide ideal a priori candidate genes that are located within these loci to investigate. GWAS have been conducted in Chinese, Korean, and European cohorts, and have identified up to 19 distinct genetic loci in, or near, known genes that are associated with PCOS [13,14,15,16,17,18,19]. Candidate gene studies are particularly useful to validate and decipher the functional impact of gene loci identified by GWAS to contextualise clinical relevance [20]. A multitude of candidate gene studies have been conducted in PCOS, identifying single nucleotide polymorphisms (SNPs) that may contribute to the genetic basis of PCOS [14]. These studies provide an effective approach for detecting genetics variants that are either causative or belong to a shared haplotype that is causative [21,22,23]. However, candidate gene studies have many common limitations, such as sample size and selection bias from confounding variables such as ancestry, diagnostic criteria, BMI, and source of participants, which can limit statistical power and result in different findings [24,25].

In all fields of science, systematic evaluations are crucial for establishing the consistency and significance of the current evidence base [24]. Such evaluations may take the form of systematic reviews of individual studies or overviews of systematic reviews. An overview of systematic reviews aims to assess the methodological quality of systematic reviews on a given topic and the consistency of evidence contained in them [26]. The aim of this overview was to systematically evaluate the current evidence regarding the genetics of PCOS from candidate gene systematic reviews and GWAS. We also explored the methodological quality of the systematic reviews to identify how future genetic studies can align the findings of candidate gene and GWAS studies by validating gene loci. A comprehensive understanding of the current evidence and its limitations is necessary to ultimately improve our understanding of the biological origins of PCOS.

## 2. Experimental Section

### 2.1. Protocol and Registration

This review was designed and reported in accordance with the Preferred Reporting Items for Systematic Reviews and Meta-Analysis (PRISMA) guidelines [27]. The protocol was registered in the international prospective register of systematic reviews PROSPERO (CRD42016052649).

### 2.2. Systematic Search of Candidate Gene Systematic Review

#### 2.2.1. Literature Search

A comprehensive database search was conducted on 17 October 2016 and updated on 5 April 2019. The following electronic databases were used to identify relevant systematic reviews: Medline in-process and other nonindexed citations (Ovid MEDLINE(R) In-Process & Other Nonindexed Citations, Ovid MEDLINE(R) Daily and Ovid MEDLINE(R) 1946 to Present), EMBASE (EBM Reviews—Cochrane Database of Systematic Reviews 2005 to 12 October 2016, EBM Reviews—ACP Journal Club 1991 to September 2016, EBM Reviews—Database of Abstracts of Reviews of Effects 1st Quarter 2015, EBM Reviews—Cochrane Central Register of Controlled Trials September 2016, EBM Reviews—Cochrane Methodology Register 3rd Quarter 2012, EBM Reviews—Health Technology Assessment 3rd Quarter 2016, EBM Reviews—NHS Economic Evaluation Database 1st Quarter 2015), and CINAHL PLUS. The search strategy for MEDLINE is documented in Appendix A. This search was modified for EMBASE and CINAHL using their subject headings instead of the MeSH subject headings. Two independent reviewers (L.J.M. or A.M.-A. (update of search) and D.H), who were not blinded to the names of investigators or sources of publication, identified and selected the systematic reviews that met the inclusion criteria.

#### 2.2.2. Eligibility Criteria and Inclusion Criteria

The Participant, Intervention, Comparison, Outcomes, and Studies (PICOs) framework was used for this Overview of Systematic Reviews (OSR) (Appendix A). Briefly, the population was women with PCOS, and the intervention was a systematic review with or without a meta-analysis with a primary focus on genetic associations. Systematic reviews not on the genetics of PCOS (e.g., focusing on assessment or treatment) were excluded and are the focus of separate OSRs [28,29]. The specific inclusion criteria for systematic reviews were a publication date from 2009 onwards, description of a search strategy containing at least key words or terms, inclusion of the number of identified and included articles, and quality appraisal of the articles. The comparison term was not applicable in this context. The outcomes included the methodology, results, and quality of each systematic review. Only articles published in English were included.

#### 2.2.3. Data Extraction

All eligible systematic reviews (SR) were examined and extracted independently by two reviewers (D.H and A.M.-A). The data extracted included information on authors, publication date, inclusion criteria, SR outcomes, number of participants in the SR, and quality of identified articles in each SR. Methodological variables specific to genetic association studies were also extracted: source of participants, whether the control group were in Hardy–Weinberg Equilibrium (HWE), and the method by which the control group was dealt with in the SR if it departed from HWE. Briefly, the HWE principle states that if control groups are healthy and therefore “disease-free”, they should be in equilibrium and genetic variation remains constant [30]. Departures from HWE can indicate a number of methodological issues including poor study design or genotyping errors. There is no consensus on which method is most appropriate to deal with deviations from HWE, but common procedures are excluding any studies that have a significant deviation from HWE before conducting meta-analysis, conducting sensitivity analysis to examine whether meta-analysis results are altered when studies containing control groups not in HWE are excluded, or correcting the pooled odds ratio [31]. For each SR the diagnostic criteria for PCOS and the criteria for defining the control group were extracted.

#### 2.2.4. Quality Assessment of Systematic Reviews

All included systematic reviews were evaluated by two independent reviewers (D.H and A.M.-A) using the Assessing the Methodological Quality of Systematic Reviews 2 (AMSTAR2) tool, which contains 16 items to appraise the methodological aspects of systematic reviews [32]. Systematic reviews were ranked as high, moderate, low, and critically low quality based on the number of critical (items 2, 4, 7, 9, 11, 13, and 15) domains missed [32]. At all stages of data extraction and quality assessment, disagreements between the two reviewers (D.H. and A.M.-A.) were discussed and resolved by consensus or arbitration (M.G.-H).

### 2.3. Semi-Structured Search of Genome-Wide Association Studies (GWAS)

GWAS are extremely informative in providing candidate genetic loci but were not included in our original systematic search and PICO. Therefore, a semi structured search was performed to identify all GWAS to date (April 2019) to provide a robust and comprehensive overview of genetics and PCOS. Keywords searched were “PCOS”, “Polycystic Ovary Syndrome”, “GWAS”, and “genome-wide association studies”. Narrative reviews and pathways analysis were excluded as this was not the focus of the paper.

## 3. Results

### 3.1. Candidate Gene Systematic Reviews

#### 3.1.1. Eligibility Assessment

The search yielded 1218 publications and 902 remained after removal of duplicates. A further 32 studies were identified from PROSPERO. Based on a priori selection criteria, screening for title or abstract identified 340 studies for assessment of full text. Of these, 143 articles were excluded due to not conducting quality assessment, not being in English, or no search terms or search strategy identified. Detailed characteristics of excluded studies are reported in Appendix A. Of the remaining 197 full-text systematic reviews, 21 were related to genetics and included in this OSR (Figure 2).

All 21 systematic reviews included a meta-analysis and were based on case-control or cross-sectional primary studies. Nineteen systematic reviews investigated SNPs [33,34,35,36,37,38,39,40,41,42,43,44,45,46,47,48,49,50,51]. The remaining two systematic reviews investigated CAG repeat length or tandem repeats respectively [52,53]. Candidate genes focused on three main aspects of PCOS pathophysiology: metabolic dysfunction, [33,35,38,39,40,41,42,43,44,48,49,50,52], imbalances in androgens and gonadotrophins [36,37,51,53], and inflammation [34,45,46,47].

#### 3.1.2. Metabolism

The genes involved in metabolic function investigated were INSR, Adiponectin, TCF7L2, IRS-1, IRS-2, Calpain-10, CY1A1, CYP11A1, PON1, DENND1A, and Insulin gene-variable number of tandem repeats (INS-VNTR) (Figure 3). In two systematic reviews, the SNP rs1801278 in IRS-1 was associated with an increased risk of PCOS when carrying the A allele, in a combined cohort of women from different ancestries [39,44]. The III allele in INS-VNTR was associated with an increased risk of PCOS compared to I allele [52], also in a combined cohort of multiple ancestries. The SNP rs4646903 in CYP1A1 and the CYP11A1 microsatellite [TTTA] n repeat polymorphism were associated with increased risk of PCOS, however not across all ancestries [41,42]. In adiponectin, the T allele in the rs1501299 SNP was shown to have a decreased risk of PCOS, however ancestry played a major role, with significant association found only in a mixed population of “East Asian” ancestry. Three SNPs in Calpain-10 were associated with increased risk of PCOS, with ancestry again playing a major role, with a significant association only found in a mixed population of “Asian” ancestry [40]. The two SNPs in PON1 were associated with increased risk of PCOS overall but, again, this was restricted to specific ancestries and diagnostic criteria [49]. Two SNPs in DENND1A (rs10818854 and rs10986105) were associated with increased risk of PCOS [48], while the SNP (rs2479106) in DENND1A was only associated in the women of mixed “Asian” ancestry [48]. In summary, the genes IRS-1, INS-VNTR, Calpain-10, PON1, CYP1A1, CYP11A1, DENND1A, and Adiponectin were found to be associated with risk of PCOS. Some of the SNPs in these genes were not consistent across ancestries indicating that future genetic studies should include larger samples sizes and investigation of other SNPs within these genes to uncover the causal SNP variant associated with PCOS in different ancestries.

#### 3.1.3. Androgens and Gonadotrophins

Four systematic reviews focused on *CYP17, FSHR, AMH, AMH receptor (AMHR II)*, and *androgen receptor (AR)* genes and their association with androgens and gonadotrophins (Figure 4). The rs6166 SNP in *FSHR* was associated with PCOS with the Asn allele showing a protective effect for PCOS, but this was only significant in a mixed population of women with European ancestry [37]. CAG repeat length polymorphism in *AR* was positively associated with plasma testosterone concentration, but not with PCOS per se [53]. There were no clear association between SNPs in *CYP17, AMH* or *AMHR* with PCOS [36,51], however, it may be possible that other SNPs within these gene loci are related to PCOS and/or specific clinical characteristics of PCOS [54]. In summary, SNPs in the genes *FSHR* and *AR* were found to be associated with risk of PCOS.

#### 3.1.4. Inflammation

Four systematic reviews focused on inflammation and investigated cytokine genes: *TNF-α*, *IL-6, IL-β*, *IL-10*, and *IL-18* (Figure 5). Of the three SNPs investigated within *TNF-α*, only one (rs1799964) was positively associated with PCOS. All four systematic reviews concurred that the C allele (rs1800795) in *IL-6* was a protective factor for PCOS risk. However, in three of the systematic reviews, when only primary studies with control groups in HWE were included, the association was no longer significant [34,45,46]. Although, the most recent systematic review [47] with the largest sample size found that this association held when controls were in HWE, indicating that rs1800795 in *IL-6* may be associated with PCOS. SNPs in the genes *IL-β*, *IL-10*, and *IL-18* were not associated with PCOS. In summary, evidence suggests that *TNF-α* (rs1800795) and *IL-6* (rs1800795) may be associated with PCOS, but larger sample sizes and further studies with appropriate control groups are required to confirm these findings.

### 3.2. Genome-Wide Association Studies

Six GWAS have been conducted to identify gene loci that are associated with PCOS. Two GWAS were conducted in women with Han Chinese ancestry (both North and South) [15,16,19], two in women with Korean ancestry [18,19], and two in women with European ancestry, determined by genetic analysis of local ancestry [13,17]. To date, only one meta-analysis of GWAS has been conducted in women with European ancestry [17]. In total, 19 genetic loci associated with risk of PCOS have been identified in three biogeographical ancestries, Korean, Han Chinese, and European. Eleven of the 19 loci are common to Han Chinese and European ancestry. Table 1 summarises the findings of the 6 GWAS. Table 2 summaries the SNPs identified from the meta-analysis including 3 novel loci [17].

#### 3.2.1. Methodological Limitations of the Systematic Reviews

Ten of the 21 included systematic reviews (48%) did not describe which PCOS diagnostic criteria they accepted in their inclusion criteria (Appendix A) [36,38,39,41,44,45,47,50,51,52]. The remaining 11 systematic reviews accepted diagnostic criteria consistent with the most inclusive Rotterdam diagnostic criteria. However, it should be noted that some of the primary studies included in these systematic reviews selectively recruited specific phenotypes (such as the National Institute of Health (NIH), or Androgen Excess PCOS (AEPCOS) phenotypes) [33,34,35,37,40,42,43,46,48,49,53] potentially indicating a bias towards the more severe phenotypes in PCOS.

Only four systematic reviews (25%) described in detail the inclusion criteria for the control groups: absence of irregular cycles, subfertility, polycystic ovarian morphology and signs of hyperandrogenism, or healthy with proven fertility (Appendix A). The remaining systematic reviews either did not describe any criteria for control group inclusion or described women in the control groups as healthy but without including any further detail.

Five systematic reviews excluded any individual studies where the control group did not conform to HWE [38,40,41,42,43,55]. One SR did not test for HWE [53]. The remaining systematic reviews (15/21) all performed sensitivity analysis of the individual studies whose control group did not conform to the HWE [34,35,36,37,39,44,45,46,52].

#### 3.2.2. Assessment of Systematic Review Quality Using the AMSTAR Tool

None of the systematic reviews met all 16 AMSTAR criteria (Appendix A) [32]. Of the 21 systematic reviews, 24% (5/21) were of high quality, indicating that the reviews provided accurate and comprehensive results, and 57% (12/21) were of moderate quality, indicating that there were weaknesses identified but were able to provide meaningful results. The remaining four systematic reviews were of low quality and should be treated with caution [32]. Of the systematic reviews relating to metabolic dysfunction, 2/13 were of high quality, 8/13 were of moderate quality, and 2/13 were of low quality. Of the systematic reviews relating imbalances in androgens and gonadotrophins, 2/4 were of moderate quality and 2/4 were of low quality. Of the systematic reviews relating to inflammation, 2/4 were of high quality and 2/4 were of moderate quality.

## 4. Discussion

This systematic overview, encompassing both candidate gene systematic reviews and GWAS, highlights gene loci that have robust associations with PCOS. These include the genes *DENN1DA, INS-VNTR*, and *INSR*, which are related to metabolic dysfunction, and *THADA* and *FSHR,* relating to imbalances in androgens and gonadotrophins pathways. However, SNPs in inflammatory genes seem unrelated to PCOS, and require additional investigation, especially in the context of obesity and PCOS.

### 4.1. Metabolic Dysfunction

Metabolic dysfunction is involved in the aetiology of PCOS [9,29]. Much research has been conducted in this area, supported by our finding that over half of the gene loci identified were concerned with metabolic dysfunction. More specifically, most studies examined genetic variants within genes that regulate insulin resistance, which is strongly implicated in the aetiology and reproductive and metabolic health consequences of PCOS [6,7]. The genetic variants from candidate gene systematic reviews, including *CYP1A1*, *CYP11A1*, *Adiponectin*, *Calpain-10*, and *PON-1*, provide promising candidate genes, but the SNPs investigated were not significant across all ancestries. This could indicate that while these genetic loci could be associated with PCOS, the genotypic variations (SNPs) within in the gene loci may vary depending on the ancestry of population being studied [56,57,58]. This is clearly illustrated by the GWAS in the *DENND1A* gene, whereby the rs10818854, rs2479106, and rs10986105 SNPs were associated with PCOS in those of Han Chinese ancestry and mixed “Asian” ancestry [15,16,48]. While in European ancestry, it was the SNP rs9696009 that was associated with risk of PCOS [17], highlighting the important role that ancestry plays in the genetics of PCOS.

SNPs in *INS-VNTR*, which are implicated in the development of T2DM, were associated in two candidate gene systematic reviews with increased risk of PCOS [40,52]. Further, the Shi et al. GWAS identified the SNP rs2059807 in *INSR* to be associated with PCOS in a Han Chinese ancestry [15]. It was then followed up in a meta-analysis by Feng et al. [33] who investigated multiple SNPs in *INSR* including the rs2059807. While they were unable to perform a meta-analysis due to insufficient data, three of the four genetic studies found a significant association with PCOS [33]. This SNP provides an ideal candidate to investigate in other ancestries, and further reinforces the importance of insulin resistance in the aetiology of PCOS.

In summary, robust candidate gene studies of the following loci are required in different ancestries: *CYP1A1*, *CYP11A1*, *Adiponectin*, *Calpain-10*, and *PON-1*. SNPs in the *INS-VNTR*, *DENND1A*, and *INSR* genes have been identified in multiple ancestries and may affect the metabolic dysfunction exhibited in PCOS. To identify the causal role of these SNPs in aetiology of PCOS, genotype–phenotype studies of SNPs in the *INS-VNTR*, *DENN1DA*, and *INSR* are warranted.

### 4.2. Dysregulation of Androgens and Gonadotrophins

Follicular arrest, menstrual dysfunction, and anovulation are commonly observed in PCOS [59,60,61], and are linked to excess androgens, FSH, and LH imbalances and elevated AMH levels. Therefore, candidate gene studies have focused on SNPs in genes such as *CYP17*, *AMH*, or *AMHR* that are associated with these pathways. From this overview, we cannot at this time establish or refute any of these candidate genes, as the quality of the current candidate gene systematic reviews were of moderate quality [37,51] or low quality [36,53].

The *FSHR* locus provides a more promising genetic loci found in the GWAS conducted by Chen et al. and Shi et al. [15,16] who examined women of Han Chinese ancestry and identified that the SNPs rs2268361 and rs2349415 were associated with PCOS. In women of European ancestry, the candidate gene systematic review conducted by Qiu et al. indicate that the SNP rs6166 in the *FSHR* gene may be the more relevant genetic variant [37]. The *FSHR* locus has also been significantly associated, in phenotypic studies, with levels of gonadotrophins (FSH and LH), indicating this may be a causal gene in the development of PCOS [62]. However, interestingly the GWAS [13,14,17] that were conducted in European ancestry did not uncover any association between the SNP rs6166 that was identified by Qiu et al. in the candidate gene systematic review. We speculate this may be due to the different diagnostic criteria used [37]. The GWAS included only women with PCOS diagnosed by the NIH criteria, and therefore potentially biased the gene loci towards the more severe phenotype of hyperandrogenism and anovulation, while the candidate gene systematic review conducted by Qiu et al. used the more inclusive Rotterdam criteria which includes the polycystic ovarian morphology (PCOM) phenotype [37]. A recent review has highlighted this conflicting relationship between the SNPs in the *FHSR* locus and risk of PCOS [63], emphasising that the SNPs may be correlated with higher basal FSH serum levels and the PCOM phenotype, rather than PCOS per se, hence the discrepancy between the GWAS findings and the candidate gene systematic review.

The loci *THADA* has been replicated across multiple ancestries and diagnostic criteria, providing another promising candidate gene in PCOS aetiology. While the functional role of *THADA* has yet to be well characterised, SNPs within the *THADA* locus have been identified as a candidate gene in T2DM, a comorbidity of PCOS [64]. Subsequent genotype–phenotype correlational analysis found that the *THADA* gene contributes to hyperandrogenism in PCOS with Han Chinese ancestry [65]. In summary, robust markers related to imbalances in androgens and gonadotrophins have been identified and future studies need to (a) identify the underlying molecular mechanism of the *THADA* and *FSHR* gene in PCOS, and (b) uncover the causal SNP variant across different ancestries and phenotypes.

### 4.3. Inflammation

Inflammation potentially acts as a link between insulin resistance and hyperandrogenism in PCOS and is associated with both [66,67]. Two candidate gene systematic reviews focused on SNPs in the *TNF-α* gene, which is a pro-inflammatory cytokine that has been associated with PCOS, ovarian function, and ovulation, and is a known mediator of insulin resistance [34,46,66]. Neither reported significant associations between the rs1800629 SNP and PCOS [34,46]. However, the SNP rs1799964 was positively associated with PCOS suggesting this may be the casual polymorphism in the *TNF-α* gene for susceptibility to PCOS [46]. Four systematic reviews examined the SNP rs1800795 in the *IL-6* gene and found a decreased risk of PCOS when carrying the C allele. Caution is warranted as when only primary studies with control groups in HWE were included, this association was no longer significant based on three of the candidate gene systematic reviews [34,45,46]. However, the most recent systematic review [47] with the largest sample size found that this association held when controls were in HWE, indicating that rs1800795 in *IL-6* may be associated with PCOS, but this requires further investigation before conclusions can be made [34,45,46,47].

The conflicting findings regarding SNPs in inflammatory genes could be explained by selection bias introduced by the source of recruitment. Compared to community based recruitment, recruitment from hospitals would lead to selection bias of the more severe hyperandrogenic-anovulatory phenotype, which has greater rates of obesity and inflammation [6]. Whether low-grade inflammation is intrinsic to PCOS or a consequence of PCOS-related obesity is contentious. Some, but not all, studies demonstrate that inflammation is independent of BMI in women with PCOS [68,69]. Unfortunately, only one of the included systematic reviews investigated the confounding influence of BMI on inflammation gene variants [34]. Obesity is known to exacerbate many of the symptoms of PCOS, and it would be prudent for future systematic reviews to investigate the role BMI may play in the association between inflammatory related gene loci and PCOS. In fact, the impact of environmental factors on PCOS is demonstrated by weight management being the first line treatment for the condition [70]. The contribution of these gene–environment interactions and epigenetics to PCOS is an emerging field, with recent findings revealing specific epigenetic reprogramming of genes involved in reproductive function in women with PCOS [71]. Therefore, it is important that future studies acknowledge and investigate the role of gene–environment interactions in the context of PCOS.

### 4.4. Recommendations for Future Genetic Analysis

We found little overlap between gene loci identified from GWAS and those identified from the candidate gene systematic reviews. This could be due to multiple reasons. Firstly, some of the gene loci identified from the candidate gene systematic reviews may have had real but weak associations in GWAS that were lost in the adjustment for multiple comparisons [72,73,74]. A second reason may be that the GWAS were not designed to be sensitive enough to detect rare genetic variants that may have a larger effect on genetic risk. Instead, the GWAS focused on common variants, and in general, these alleles may only have a small effect on genetic risk [74,75]. This was recently shown in a family-based association study that identified multiple rare genetic variants in *DENN1DA* that would have been unable to be detected in GWAS, and these rare SNPs were found to contribute to PCOS and, in particular, to hormonal imbalances [76]. GWAS have identified less than ∽10% of PCOS heritability, while twin and family studies indicate that heritability may be up to ∽70%. This emphasises that a combined approach of GWAS, candidate gene association, and family-based studies are required to fully elucidate the genetic contribution to the origins of PCOS [77].

Future genetic studies should consider performing Phenome-Wide Association Studies (PheWAS) which examine many different phenotypes to see which, if any, are associated with a given genetic variant [78]. The findings of this overview provide potential pathways that could be used in PheWAS to determine the functional relevance of the identified genes, and secondly, to explore associations between genetic polymorphisms and different PCOS phenotypes. The importance of this has been recently highlighted in a GWAS meta-analysis where it was identified that the SNP rs804279 in the *GATA4/NEIL2* gene loci was strongly associated with the NIH diagnostic criteria, which encompasses only hyperandrogenic PCOS phenotypes, compared to the Rotterdam criteria, which also encompasses the non-hyperandrogenic PCOS phenotypes [17]. Rigorous reporting and examination of the differing diagnostic criteria, and therefore PCOS phenotypes, will be particularly important to both genetic studies and PheWAS to elucidate whether the four phenotypes of PCOS may have different molecular origins [6].

This overview is a timely reminder of important methodological considerations for future genetic studies in PCOS and in complex diseases more generally. Genetic studies need to be more meticulous in the reporting of ancestry to determine genetic variation in humans. Many studies used the term “race/ethnicity” which is not appropriate in a genetic study, as commonly used ethnicity terms including Caucasian, Asian, White, African, or Latino are poor predictors of human genetic variation or similarity [58,79]. Instead it would be more accurate to calculate ancestry or the geographical origins of individual participant’s ancestry (biogeographical ancestry) to uncover genetic variation in PCOS. Ancestry Informative Markers (AIMs) can estimate biogeographical ancestry, especially those of mixed ancestral background, and provide an improved understanding of the impact of genes in PCOS [75,79].

Criteria for control groups need to be clearly defined, as most of the systematic reviews simply stated that they included healthy women or did not define the relevant inclusion criteria. This may affect the strength of association [75,80], as suggested by two systematic reviews reported in this OSR that included the same primary studies but came to different conclusions [38,43]. While Ramos et al. (2015) excluded any controls from their meta-analysis that were not considered healthy, Shen et al. (2014) did not describe the inclusion criteria for the control group, therefore it is difficult to compare the meta-analyses to understand where this variation in findings originated from. Another contentious issue is whether to include or exclude individual primary studies whose control groups did not conform to the Hardy–Weinberg Equilibrium (HWE) [31,81]. We note a variety of methods were used to deal with the primary studies that departed from HWE, and some of the systematic reviews did not consider this issue at all. Almost all systematic reviews acknowledged they were limited by a small sample size, highlighting the need for larger primary studies and systematic reviews, with the candidate genes informed by family-based studies and GWAS. Although most systematic reviews were at low risk of bias, there was a lack of consistent methodological rigour regarding clear definitions of cases and controls, ancestry, and the differences in dealing with deviations from HWE, which should be addressed to progress our knowledge of the role of genetics in the aetiology of PCOS.

## 5. Conclusions

This overview of systematic reviews and GWAS identified several PCOS candidate gene loci that are located in the neuroendocrine, metabolic, and reproductive pathways. Additionally, we described the limitations and important methodological considerations that should inform and complement future genetic studies. We have provided a comprehensive catalogue of gene loci, with work now required to identify causal variants and functional relevance to the biological origins and established pathophysiology of PCOS.

## Figures and Tables

**Figure 1 jcm-08-01606-f001:**
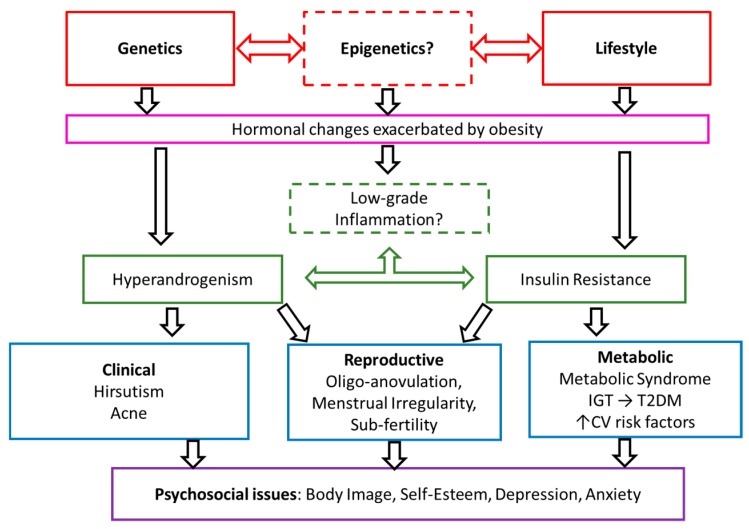
Proposed pathophysiology and features of Polycystic Ovary Syndrome (PCOS). Adapted and reproduced with permission [2]. CV, cardiovascular; IGT, impaired glucose tolerance; TD2M, type 2 diabetes.

**Figure 2 jcm-08-01606-f002:**
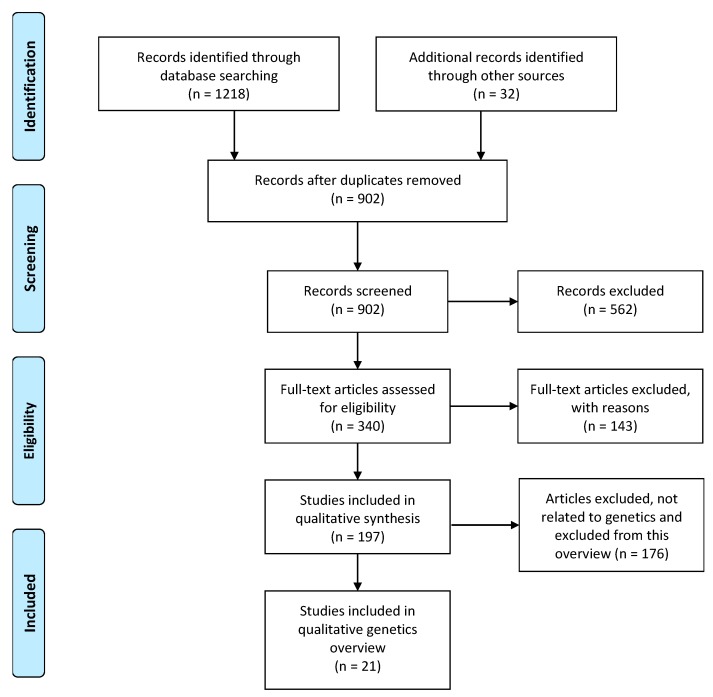
Identification and selection of systematic reviews of genetics and polycystic ovary syndrome.

**Figure 3 jcm-08-01606-f003:**
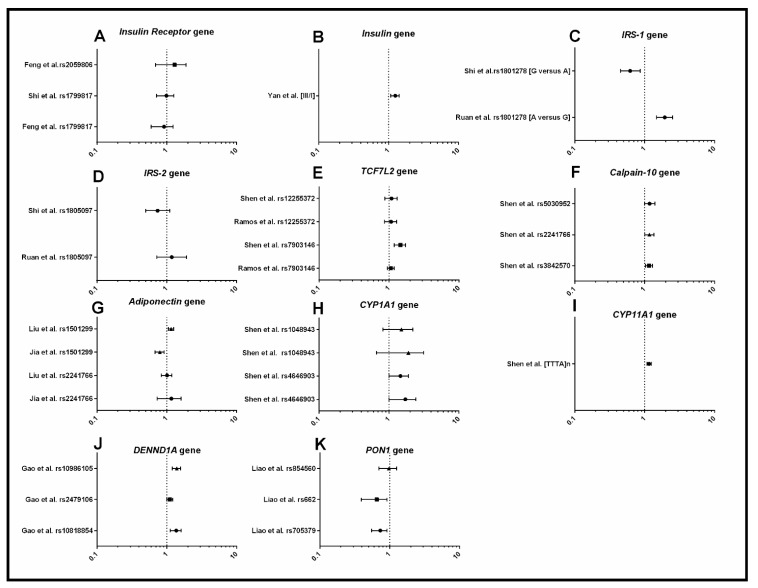
Forest plots representing the association between metabolic SNPs and PCOS under the allele model. (**A**) Insulin receptor; (**B**) Insulin gene variable number of tandem repeats (INS-VNTR); (**C**) Insulin Receptor Substrate-1 (IRS-1); (**D**) Insulin Receptor Substrate-2 (IRS-2); (**E**) Transcription Factor 7-Like 2 (TCF7L2); (**F**) Calpain-10; (**G**) Adiponectin; (**H**) Cytochrome P450 Family 1 Subfamily A Member 1 (CYP1A1); (**I**) Cytochrome P450 Family 11 Subfamily A Member 1 (CYP11A1); (**J**) DENN domain containing 1A (DENND1A); (**K**) Paraoxonase 1 (PON1). The circle represents the odds ratio (OR) and the horizontal lines are the 95% confidence intervals (CI).

**Figure 4 jcm-08-01606-f004:**
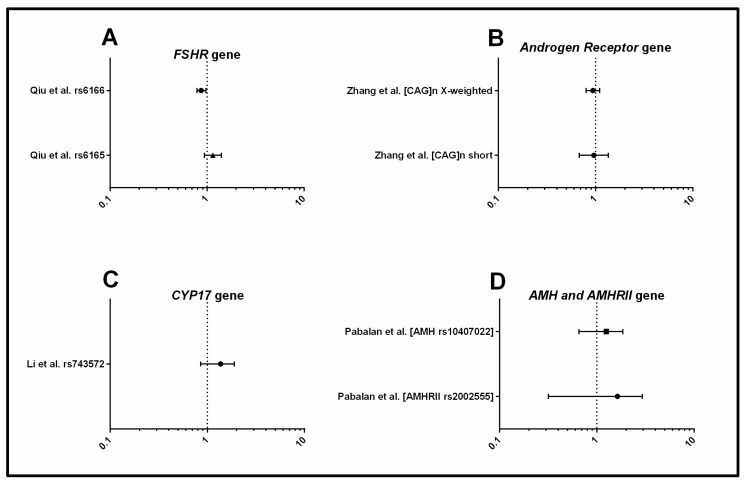
Forest plots representing the association between metabolic SNPs and PCOS under the allele model. (**A**) Follicle Stimulating Hormone Receptor (FSHR); (**B**) Androgen Receptor (AR); (**C**) Cytochrome P450 Family 17 Subfamily A Member 1 (CYP17); (**D**) Anti-müllerian hormone (AMH); Anti-müllerian hormone receptor II (AMHRII). The circle represents the odds ratio (OR) and the horizontal lines are the 95% confidence intervals (CI).

**Figure 5 jcm-08-01606-f005:**
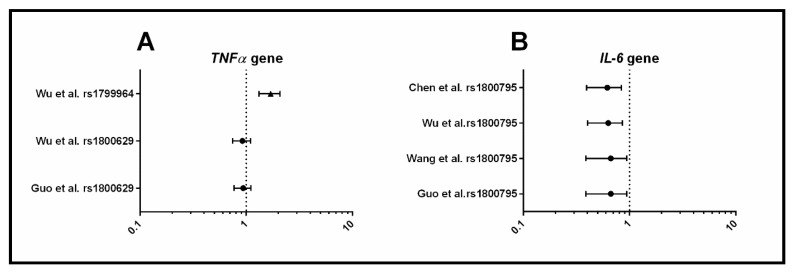
Forest plots representing the association between metabolic SNPs and PCOS under the allele model. (**A**) *TNF-α*, Tumor Necrosis Factor-Alpha; (**B**) *IL*, Interleukin. The circle represents the odds ratio (OR) and the horizontal lines are the 95% confidence intervals (CI).

**Table 1 jcm-08-01606-t001:** SNPs identified by Genome-Wide Association Studies (GWAS) in Polycystic Ovary Syndrome (PCOS). NIH, National Institute of Health.

Study	Diagnostic Criteria	Gene Locus	SNPs	Nearest Gene
Chen et al., 2011 [11]	Rotterdam	2p16.3	rs13405728	*LHCGR, STON1-GTF2A1L*
		2p21	rs12468394	*THADA*
			rs13429458	
			rs12478601	
		9q33.3	rs10818854	*DENND1A*
			rs2479106	
			rs10986105	
Shi et al., 2012 [15]	Rotterdam	2p16.3	rs13405728	*LHCGR, STON1-GTF2A1L*
		2p16.3	rs2268361	*FSHR*
			rs2349415	
		2p21	rs12468394	*THADA*
			rs13429458	
			rs12478601	
		9q33.3	rs10818854	*DENND1A*
			rs2479106	
			rs10986105	
		9q22.32	rs4385527	*C9orf3*
			rs3802457	
		11q22.1	rs18974116	*YAP1*
		12q13.2	rs705702	*RAB5B, SUOX*
		12q14.3	rs2272046	*HMGA2*
		16q12.1	rs4784165	*TOX3*
		19p13.3	rs2059807	*INSR*
		20q13.2	rs6022786	*SUMO1P1*
Lee et al., 2015 [19]	Rotterdam	8q24.2	rs10505648	*KHDRBS3, LINC02055*
Hwang et al., 2012 [18]	Rotterdam	12p12.2	rs10841843	*GYS2*
			rs6487237	
			rs7485509	
Hayes et al., 2015 [13]	NIH	8p32.1	rs804279	*GATA4, NEIL2*
		9q22.32	rs10993397	*C9orf3*
		11p14.1	rs11031006	*ARL14EP, FSHB*
Day et al., 2015 [14]	NIH	2q.34	rs1351592	*ERBB4*
		11q22.1	rs11225154	*YAP1*
		2q21	rs7563201	*THADA*
		11p14.1	rs11031006	*FSHB*
		5q31.1	rs13164856	*RAD50*
		12q21.2	rs1275468	*KRR1*

**Table 2 jcm-08-01606-t002:** SNPs identified from a GWAS meta-analysis in women of European biogeographical ancestry.

Study	Diagnostic Criteria	Gene Locus	SNPs	Nearest Gene
Day et al. [17]	Rotterdam NIH	2q21	rs7563201	*THADA **
		2q.34	rs2178575	*ERBB4*
		5q31.1	rs13164856	*IRF1/RAD50*
		8p32.1	rs804279	*GATA4/NEIL2*
		9p24.1	rs10739076	*PLGRKT*
		9q22.32	rs7864171	*C9orf3 **
		9q33.3	rs9696009	*DENND1A **
		11p14.1	rs11031005	*ARL14EP/FSHB*
		11q22.1	rs11225154	*YAP1 **
		11q23.2	rs1784692	*ZBTB16*
		12q13.2	rs2271194	*ERBB3/RAB5B*
		12q21.2	rs1795379	*KRR1*
		16q12.1	rs8043701	*TOX3 **
		20q11.21	rs853854	*MAPRE1*

* Gene loci that have been found in common in women of Han Chinese ancestry.

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
