# Peer review of "The Genetics of Polycystic Ovary Syndrome: An Overview of Candidate Gene Systematic Reviews and Genome-Wide Association Studies"

_jcm, 2019, doi:10.3390/jcm8101606_

Round 1

Reviewer 1 Report

English language is adequate

Topic is of great scientific interest, due to exact mechanisms are not yet fully understood, therefore this article explains lots of aspect of this topic.

Meta-analysis is well organized and materials and methods give strength to the study.

Results are well presented.

Discussion is well written and future perspectives in genetics and PCOS give a great aspect from this study.

Reviewer 2 Report

Very useful article to study the important genes playing a role in the cause of PCOS. And their methodology looks good to me. Their way is the best way to find what is close to the truth. However it must always keep in mind that the genetic findings must be in accordance with the phenotype, the study of the clinical symptoms, and the authors also mention that.

The abstract would be much better if it is divided in chapters like intro, methodology, findings or results and conclusion. What I absolutely miss is that the findings as described in their summaries of Metabolism line 188-191 and Androgen and gonadotrophins line 2017 and 2018 and Inflammation line 223, 224, 225. And not in the abstract.

The genes the authors mention  in the chapter of Metabolism for instance Cyp1A1 and PON1 are well known to be related to general environmental pollution for instnace with dioxins and with pesticides like chlorpyrifos. But the authors say that they don’t want to address that. That is ok, but some remarks about possible gene-environment interaction amd epigenetics might have been good.

About the chapter on Inflammation:

I wonder if it is useful to look also for the HLA-type of the patients. That might bring closer also to genetics.

In 236 and 237 and 238 the biogeographical ancestries are mentioned. I miss the Hindustani, are they taken together with the Europeans? I remember a difference in HLA-type between descendants of Hindustani with gestational diabetes with significantly more DR2 in relation to Europeans. Anyhow this is not a critic, but just a remark.

I think the article is very useful for studying the genetic origins of PCOS and I am in favour of accepting it after restruction of the abstract.
